# What Do We Know So Far about Ofatumumab for Relapsing Multiple Sclerosis? A Meta-Analytical Study

**DOI:** 10.3390/healthcare10112199

**Published:** 2022-11-02

**Authors:** Hafiza Munazza Taj, Maryam Talib, Sania Siddiqa, Azza Sarfraz, Zouina Sarfraz, Karla Robles-Velasco, Ivan Cherrez-Ojeda

**Affiliations:** 1Research and Publications, Fatima Jinnah Medical University, Lahore 54000, Pakistan; 2Research, Services Institute of Medical Sciences, Lahore 54000, Pakistan; 3Research, Ameer-ud-Din Medical College, Lahore 54000, Pakistan; 4Pediatrics and Child Health, The Aga Khan University, Karachi 74000, Pakistan; 5School of Health, Universidad de Especialidades Espíritu Santo, Samborondón 092301, Ecuador; 6Allergy, Immunology & Pulmonary Medicine, Universidad Espíritu Santo, Samborondón 092301, Ecuador

**Keywords:** ofatumumab, B cells, relapsing multiple sclerosis, clinical trials, meta analysis, neurodegeneration

## Abstract

Ofatumumab is a monoclonal antibody that reduces the level of B cells that alter the progression of relapsing multiple sclerosis. Originally approved by the Food and Drug Administration (FDA) in August 2020, this meta-analysis determines the outcomes of four randomized controlled trials (RCTs) for endline outcomes of Gadolinium-enhancing T1 lesions on MRI scans reported as Cohen’s d and relapse rate reported as risk ratio. All four RCTs reported favorable findings of gadolinium-enhancing T1 lesions (Cohen’s d = −0.44, *p* < 0.00001). The relapse rate was reduced by 46% post ofatumumab administration (RR = 0.54, *p* < 0.00001). With 14 ongoing trials in this area, more data is required to consolidate our findings.

## 1. Introduction

Ofatumumab, a monoclonal antibody, works by reducing the level of B cells which contribute to the development and progression of MS [1]. Ofatumumab, a B-cell depleting medication delivered via subcutaneous injection [2], was approved by the FDA in August 2020 for adults with relapsing forms of MS, including clinically isolated syndrome, relapsing–remitting disease, and active secondary progressive MS [3]. In MS, B cells are posited to act via the antibody production and antigen presentation system to activate T cells; they are also a vital source of pro-inflammatory cytokines, which in unison orchestrate inflammatory infiltration in the central nervous system [4]. MS is an incurable disease that affects an estimated 2.8 million individuals across the world. While the exact mechanism with which ofatumumab works is not currently known, it is understood that the FAB portion of the drug inhibits the transmembrane phosphoprotein—CD20; this region is different compared to other anti-CD20 antibodies previously used for MS [1]. B-cell lysis associated with Ofatumumab correlates to completement-dependent cytotoxicity and antibody-dependent cell-mediated cytotoxicity [4]; this is unlike the mechanism seen in other treatments, including ocrelizumab and rituximab, which only bind to the large extracellular loops of the CD20 antigen; ofatumumab binds to both the small and large extracellular loops [4,5]. The key pharmacological properties of ofatumumab are, firstly, depleting the B cells, secondly, repleting the B cells, and thirdly, immunogenicity [1].

It is pertinent to consolidate existing therapeutic options to limit progression and disability associated with disease. With monoclonal antibodies or teriflunomide available as modalities to prevent relapsing MS, ofatumumab may be an emerging and viable therapy for patients considering self-administration once per month [6]. The applicability of ofatumumab has been evaluated in an online-based survey where 250 neurologists were questioned on their attitudes towards the therapy among other questions [7]. The key findings of the survey included 90% positive responses towards early use of the therapy [7]. Moreover, the neurologists believed that reduction of relapses, mode of administration and application intervals (i.e., daily use versus monthly use) are extremely imperative aspects of relapsing MS treatment.

It is all the more important to emphasize the importance of innovative relapsing MS therapies for treatment-naïve patients. Gadolinium-enhancing T1 lesions are critical in reflecting active disease and are critical for monitoring in MS. In general, relapses of disease are the defining feature of relapsing MS—the most prevalent MS phenotype [8]. While relapses are utilized in diagnoses, the value of relapse rates is seen with the high risk of association with incomplete remission, which can lead to residual disability [8,9]. Moreover, relapse frequency early in the course of MS has strong correlations with long-term disability [9]. The objectives of this report are to quantify, firstly, the outcomes of gadolinium-enhancing T1 lesions on MRI scan at treatment endline, and secondly, the relapse rate at treatment on usage of ofatumumab at endpoint. This meta-analysis will quantify the use of ofatumumab for relapsing MS using objective-based evidence from randomized controlled trials.

## 2. Methods

PubMed, Cochrane CENTRAL, Embase, and ClinicalTrials.Gov were systematically searched for randomized controlled trials (RCTs) with MeSH terms including “Ofatumumab” and “Relapsing Multiple Sclerosis”. The full keyword string for PubMed is as follows: Ofatumumab: “ofatumumab” [Supplementary Concept] OR “ofatumumab” [All Fields]; Relapsing: “recurrence” [MeSH Terms] OR “recurrence” [All Fields] OR “relapse” [All Fields] OR “relapses” [All Fields] OR “relapsing” [All Fields] OR “relapsed” [All Fields] OR “relapse” [All Fields] OR “relapsers” [All Fields]; Multiple Sclerosis: “multiple sclerosis” [MeSH Terms] OR (“multiple” [All Fields] AND “sclerosis” [All Fields]) OR “multiple sclerosis” [All Fields]. The databases were searched from inception until 5 September 2022. No language restrictions were applied; any non-English language study, if identified, was to be translated into English using Google Translate.

The inclusion criteria covered RCTs only enrolling adult patients aged 18 and above, of any gender, with definite diagnosis of relapsing MS as per the study-defined criteria. The participants were required to be intervened with ofatumumab, with placebo groups as comparators. Observational studies, case reports/series, systematic reviews/meta-analyses, brief reports, and letters to editors were excluded from this study.

In the screening phase, the titles and abstracts of shortlisted studies from the databases were screened independently by two reviewers (Z.S. and A.S.). In case of any disagreements, a third reviewer resolved any issues and reached a consensus (I.C.O). Figure 1 depicts the study selection process in accordance with the Preferred Reporting Items for Systematic Reviews and Meta-Analyses (PRISMA) [10]. The data software EndNote X9 (Clarivate, London, UK) was used to omit any duplicates during the selection process and for storage of bibliographic entries. The kappa score was determined for inter-rater reliability as a measure of agreement between the two independent raters using the Statistical Package for Social Sciences (SPSS, v24).

All quantitative data was collated into a data sheet by all authors for (i) mean number and standard deviation (SD) of Gd-enhancing T1 lesions per MRI scan, and (ii) proportion of participants presenting with relapse in intervention and placebo groups. The primary aim of this meta-analysis was to ascertain the effect size (standardized mean difference), reported as Cohen’s d, and comparing the differences in mean Gd-enhancing T1 lesions per MRI scan among intervention and control groups. The secondary aim was to determine the risk ratio of relapses in disease. Cohen’s d and risk ratio (RR) were computed applying 95% confidence intervals and were set to a significance level of less than 0.05. The findings were presented as forest plots along with the *p*-values. The I^2^ index was utilized to calculate the heterogeneity among the included studies. All statistical tests were conducted in Review Manager 5.4.1 (RevMan, Cochrane).

The included RCTs were assessed for quality using version 2 of the Cochrane risk-of-bias tool for randomized trials (RoB 2). The RoB 2 tool assesses five domains comprised of the following: (1) bias arising from the randomization process, (2) bias due to deviations from intended interventions, (3) bias due to missing outcome data, (4) bias in the measurement of the outcome, and (5) bias in the selection of the reported result. The domain-based judgements were reported as (1) low risk of bias, (2) some concerns, and (3) high risk of bias. The results were reported as a traffic light plot of bias assessment and the weighted summary plot of overall domain-based type of bias.

## 3. Results

Of the 726 studies located, post appraisal by all authors, a total of four RCTs were selected for inclusion (Figure 1). We included two randomized, double-blind, double-dummy, parallel-group phase 3 trials (ASCLEPIOS I and II) [11] and two phase 2, randomized double-blind, placebo-controlled parallel-group trials (MIRROR Trial and APOLITOS Study), with a total of 2177 participants (Ofatumumab = 1153; Control = 1024) evaluating the efficacy of ofatumumab in patients with relapsing MS [12,13]. Gd-enhancing T1 lesions’ Cohen’s d had a medium effect direction in favor of ofatumumab as compared to control (Cohen’s d = −0.44, 95% CI = −0.56, −0.31; *p* < 0.00001, I^2^ = 31%) (Figure 2). On assessing the relapse rate among the intervention and control groups, it was ascertained that the risk of relapse was reduced by 46% among those intervened with ofatumumab compared to the control group (RR = 0.54, 95% CI = 0.46, 0.63; *p* < 0.00001, I^2^ = 0%). The data of these four trials are appended in Table 1 and Table 2.

Currently, there are 14 ongoing clinical trials, of which six are in phase 3 and seven are in phase 4 of testing—with an enrollment of 5465 participants addressing the efficacy and safety profile for relapsing MS (Table 3).

On noting the bias arising from the randomization process, only one trial had some concerns, whereas three trials had low concerns. On calibrating the biases arising due to deviations from the intended interventions, all four trials had low concerns. On assessing biases due to missing outcome data, all trials had low concerns, whereas assessment of bias in the measurement of the outcome resulted in one trial with some concerns and three with low concerns. For bias in the reported result, all four trials had low concerns. Overall, all four trials had low concerns (Figure 3).

## 4. Discussion

On noting the efficacy of ofatumumab, the annualized relapse rate (ARR) for individuals with relapsing MS was decreased by a mean of 54.5% among both ASCLEPIOS I and II trials [11]. Moreover, the reduction in gadolinium-enhancing T1 lesions was noted at a relative rate of 97% and 94% in both trials, respectively (*p* < 0.001) [11]. In ASCLEPIOS I and II, the safety profile was similar to that displayed by teriflunomide—a drug which has shown 79% individuals with relapsing MS have remained free of disability progression, as compared to 80% with placebo; however, teriflunomide has not always been known to achieve statistically significant reduction in the risk of sustained disability progression [11]. The findings of this meta-analysis collate pooled evidence depicting the comparability of frequency of serious infections and neoplasms being comparable between treatment and control groups. In both ASCLPEIOS I and II, the infection-related adverse reactions have been 20.2% versus 15% among ofatumumab and teriflunomide/control groups [11]. The most commonly reported adverse events are headache, injection-site reaction, nasopharyngitis, urinary tract infection and upper respiratory tract infection [11].

The MIRROR trial explored minimally effective doses of ofatumumab to identify a potential treatment for relapsing MS. In the efficacy analysis, the treatment significantly reduced new GdE lesions by 65% as compared to placebo [12]. It ought to be noted that the endline assessment of MRI outcomes were included in this meta-analysis that provide more accurate efficacy measures. Currently approved anti-CD20 treatment for relapsing MS has shown either complete or near complete depletion of circulating B cells, although it is unclear if this is essential for high efficacy outcomes [12]. In the MIRROR trial, ofatumumab led to rapid-dose dependent B-cell depletion, where 60 mg dosage over 12 weeks provided maximum benefit; a higher dosage did not provide more robust treatment effects [12]. The tolerability and adverse event findings in the MIRROR trial were comparable and any symptomatology resolved within one day of onset [12].

Finally, the phase 2 APOLITOS study ascertained that ofatumumab reduced gadolinium-enhancing T1 lesions by 93.6% compared to placebo, and the findings were consistent across regions (Japan and Russia) [13]. The extension part had comparable benefits as well; however, the adverse events were determined to be lower with ofatumumab (69.8%) as compared to placebo (81%). Injection-related adverse reactions were the most common, and no opportunistic infections, deaths, or malignancies were reported. Safety findings were also consistent with pivotal trials [13].

## 5. Conclusions

The prospect of an effective subcutaneous B-cell targeted therapy, ofatumumab, increases the possibility of self-administration and improvement over available intravenous administration medications. However, while there is demonstrated convenience of usability and optimization of healthcare resources with this intervention, it remains to be seen whether the ongoing trials support the repletion of B cells achieved with ofatumumab, and whether the therapy will emerge with favorable safety and efficacy profiles. Our findings support the favorable effects of the administration of ofatumumab, subcutaneously in the controlled trial setting.

## Figures and Tables

**Figure 1 healthcare-10-02199-f001:**
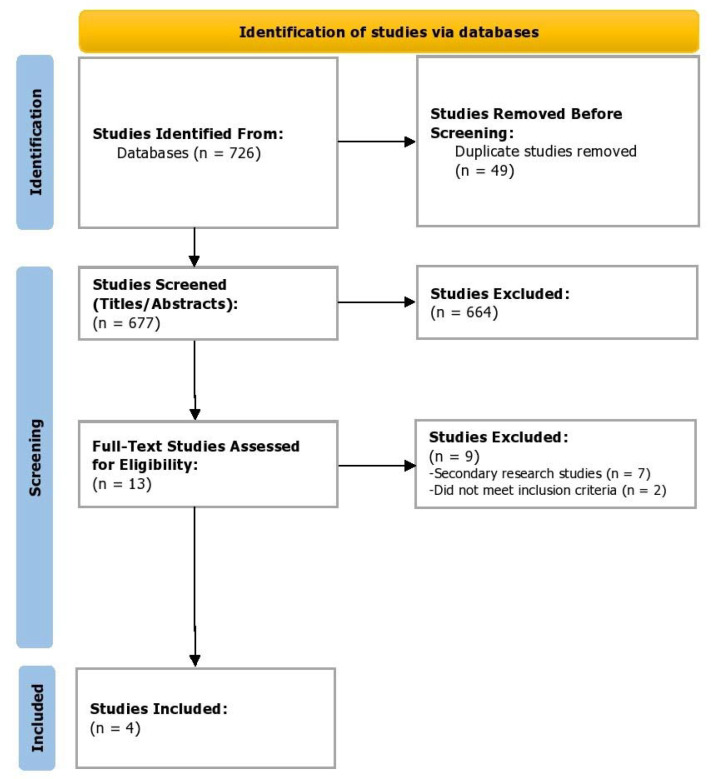
PRISMA flowchart depicting the study selection process.

**Figure 2 healthcare-10-02199-f002:**
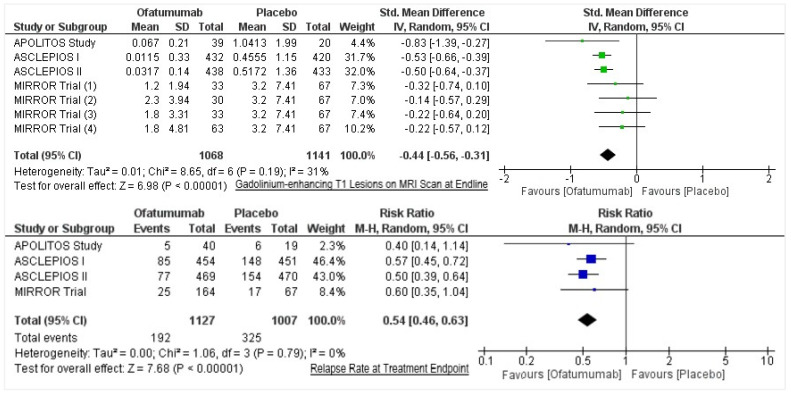
Forest plots for gadolinium-enhancing T1 lesions on MRI scan at endline and relapse rate at treatment endpoint.

**Figure 3 healthcare-10-02199-f003:**
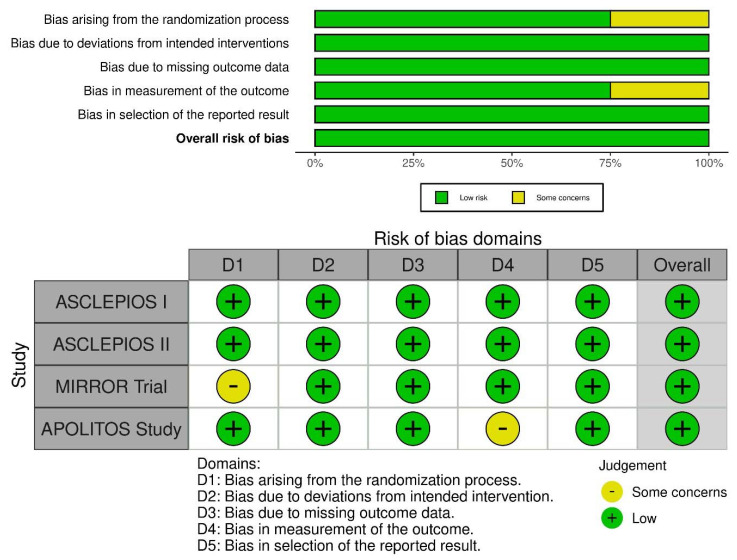
Risk of bias assessment of included RCTs using the ROB-2 tool. The weighted summary plot depicts the overall type of bias encountered in all studies. The traffic light plot represents study-by-study bias assessment.

**Table 1 healthcare-10-02199-t001:** Study characteristics of the included trials (N = 4) in the meta-analysis.

Author, Year	Hauser et al. (2020) [11]		Bar-or et al. (2018) [12]	Kira et al. (2021) [13]
Names	ASCLEPIOS I	ASCLEPIOS II	MIRROR Trial (OMS112831)	APOLITOS Study
Study type	Randomized, double-blind, double-dummy, parallel-group study (NCT02792218)	Randomized, double-blind, double-dummy, parallel-group study (NCT02792231)	Phase 2, randomized, double-blind, placebo-controlled, parallel-group, dose-ranging study (NCT01457924)	Phase 2, randomized, double-blind, placebo-controlled, parallel-group, multicenter study (NCT03249714)
Arms	Ofatumumab vs. active comparator (Teriflunomide)/placebo on days 1, 7, 14, week 4 and every 4 weeks thereafter		Ofatumumab every 4 or 12 weeks vs. placebo for a 24 week period	Randomized (2:1) to ofatumumab or matching placebo
Intervention arm sample size	N = 465	N = 481	N = 164	N = 43
Control arm sample size	N = 462	N = 474	N = 67	N = 21
Duration of intervention and follow-up	Up to 2.5 years		6 months, with follow-up until 12 months	6 months, with follow-up until 12 months
Country	Argentina, Australia, Belgium, Bulgaria, Canada, Croatia, Czechia, Denmark, Estonia, France, Germany, Greece, Hungary, India, Israel, Italy, Mexico, Netherlands, Poland, Puerto Rico, Russian Federation, Slovakia, Spain, Sweden, Switzerland, Thailand, Turkey, United Kingdom, United States	Argentina, Australia, Austria, Belgium, Bulgaria, Canada, Croatia, Czechia, Finland, France, Germany, Hungary, India, Italy, Latvia, Lithuania, Mexico, Norway, Peru, Poland, Portugal, Russian Federation, Slovakia, South Africa, Spain, Switzerland, Taiwan, Turkey, United Kingdom, United States	Bulgaria, Canada, Czechia, Denmark, Germany, Italy, Netherlands, Norway, Russian Federation, Spain, United States	Japan and Russia
Year	20 September 2016–5 July 2019	26 August 2016–10 July 2019	1 November 2011–23 August 2013	15 March 2018–26 December 2019
Primary outcome measure	To identify the annualized relapse rate in different arms at baseline up to 2.5 years		To identify the cumulative number of new gadolinium-enhancing (GdE) brain lesions at week 12 (based on T1-weighted MRI scans at weeks 4, 8, and 12)	To identify the number of gadolinium-enhancing T1 lesions per scan over 24 weeks
Secondary outcome measure(s)	To identify the (i) disability worsening at 3 months or 6 months, (ii) disability improvement at 6 months, (iii) the number of gadolinium-enhancing lesions per T1-weighted magnetic resonance imaging scan, (iv) the annualized rate of new or enlarging lesions on T2-weighted MRI, (v) serum neurofilament light chain levels at month 3, and (vi) change in brain volume		To identify the (i) cumulative number of new GdE lesions at week 24, (ii) cumulative number and total volume of new and new plus persisting GdE lesions, (iii) new and/or newly enlarging T2 lesions, and (iv) T1-hypointense lesions at weeks 12 and 24	To identify the (i) number of gadolinium-enhancing T1 lesions per MRI Scan (Japan vs. Non-Japan), (ii) number of new or enlarging T2 lesions on MRI Scans (annualized T2 lesion rate), (iii) annualized relapse rate (ARR), (iv) pharmacokinetic (PK) concentrations of ofatumumab, (v) B-cell counts
Key inclusion criteria	(I) Diagnosis of MS; (II) Relapsing MS: relapsing–remitting MS (RRMS) or secondary progressive MS (SPMS); (III) At least one relapse during the previous 1 year or two relapses during the previous 2 years, or a positive gadolinium-enhancing MRI scan in the previous year; (IV) Expanded disability status scale (EDSS) score of 0 to 5.5; (V) Neurologically stable within 1 month prior to randomization		(I) Definite diagnosis of MS according to the 2010 revisions of the McDonald diagnostic criteria for MS; (II) No manifestation of another type of MS other than RRMS; (III) Expanded Disability Status Scale (EDSS) score of 0–5.5 (inclusive) at screening; (IV) Neurologically stable with no evidence of relapse for at least 30 days prior	(I) Diagnosis of multiple sclerosis (MS); (II) Relapsing MS; (III) At least one appearance of a new neurological abnormality or worsening of pre-existing neurological abnormality during the previous 2 years prior to screening AND an MRI activity (Gd-enhancing T1 lesions or new or enlarging T2 lesions) in the brain during the previous 1 year prior to randomization; (IV) EDSS score of 0 to 5.5
Intervention	Ofatumumab on days 1, 7, 14, week 4 and every 4 weeks thereafter OR placebo taken orally once daily OR teriflunomide taken once daily OR matching placebo of ofatumumab on days 1, 7, 14, week 4 and every 4 weeks thereafter		One dose of ofatumumab 3 mg over 24 weeks OR two doses of ofatumumab 3 mg over 24 weeks OR two doses of ofatumumab 30 mg over 24 weeks OR conditioning dose of ofatumumab 3 mg at randomization, two doses of ofatumumab 30 mg over 24 weeks OR two doses of ofatumumab 60 mg over 24 weeks OR conditioning dose of ofatumumab 3 mg at randomization, two doses of ofatumumab 60 mg over 24 weeks OR six doses of ofatumumab 60 mg over 24 weeks	Ofatumumab on Days 1,7, 14 and every 4 weeks for 24 weeks; all extension patients received dose every 4 weeks up to week 48.

**Table 2 healthcare-10-02199-t002:** Patient characteristics and outcomes of the included trials (N = 4) in the meta-analysis.

Author, Year	Hauser et al. (2020) [11]		Bar-or et al. (2018) [12]	Kira et al. (2021) [13]
Names	ASCLEPIOS I	ASCLEPIOS II	MIRROR Trial (OMS112831)	APOLITOS Study
Dose and Mode of Administration	Ofatumumab 20 mg pre-filled syringes for subcutaneous injection OR placebo capsule OR teriflunomide 14 mg oral capsule OR matching placebo of ofatumumab subcutaneous injections		3 mg OR 30 mg OR 60 mg, subcutaneous	20 mg, (50 mg/mL, 0.4 mL content), subcutaneous
Age (Mean SD)	38.9 (8.77) vs. 37.8 (8.95)	38.0 (9.28) vs. 38.2 (9.47)	37.2 (9.39) vs. 37.7 (9.38)	35.0 (9.49) vs. 35.5 (8.93)
Female (n, %)	318/465 (68.4%) vs. 317/470 (68.6%)	319 (66.3%) vs. 319 (67.3%)	109/164 vs. 46/67	36 vs. 19
Race (n, %)	Asian: 21 vs. 19; Black or African American: 15 vs. 20; White: 411 vs. 412; Other/Unknown: 24 vs. 14	Asian: 21 vs. 19; Black or African American: 13 vs. 18; White: 418 vs. 417; Other/Unknown: 29 vs. 20	White: 160/164 vs. 65/67	Asian: 21 vs. 22; White: 22 vs. 10
Gadolinium-enhancing T1 Lesions on MRI Scan at Endline (Mean, SD)	0.0115 (SD = 0.33) (N = 432) vs. 0.4555 (SD = 1.15) (N = 420)	0.0317 (SD = 0.14) (N = 438) vs. 0.5172 (SD = 1.36) (N = 433)	[Ofatumumab 3 mg q12w = 1.2 (1.94) N = 33; Ofatumumab 30 mg q12w = 2.3 (3.94) N = 30; Ofatumumab 60 mg q12w = 1.8 (3.31) N = 33; Ofatumumab 60 mg q4w = 1.8 (4.81) N = 63] vs. Placebo/Ofatumumab 3 mg = 3.2 (7.41) N = 67	0.0670 (SD = 0.21) (N = 39) vs. 1.0413 (SD = 1.99) (N = 20)
Relapse Rate at Treatment Endpoint (n/N)	85/454 vs. 148/451	77/469 vs. 154/470	25/164 vs. 17/67	5/40 vs. 6/19

**Table 3 healthcare-10-02199-t003:** Overview of ongoing clinical trials.

No.	NCT Number	Title	Phase	Enrollment
1	NCT05199571	Study of Efficacy and Safety of Ofatumumab in Relapsing Multiple Sclerosis (RMS) Patients in China	Phase 4	100
2	NCT04486716	A Single-Arm Study Evaluating the Efficacy, Safety and Tolerability of Ofatumumab in Patients with Relapsing Multiple Sclerosis (OLIKOS)	Phase 3	100
3	NCT03650114	Long-Term Safety, Tolerability and Effectiveness Study of Ofatumumab in Patients with Relapsing MS (ALITHIOS)	Phase 3	2010
4	NCT04353492	An Open-Label Study Evaluating Ofatumumab Treatment Effectiveness and PROs in Subjects with RMS Transitioning from Fumarate-based RMS Approved Therapies or Fingolimod to Ofatumumab (ARTIOS)	Phase 3	555
5	NCT04510220	Nine-Month Study to Assess the Efficacy of Ofatumumab on Microglia in Patients with Relapsing Forms of Multiple Sclerosis	Phase 3	10
6	NCT04667117	A Multicenter Study to Assess Response to Influenza Vaccine in Multiple Sclerosis Participants Treated with Ofatumumab	Phase 4	66
7	NCT04869358	Exploring the Immune Response to SARS-CoV-2 COVID-19 Vaccines in Patients with Relapsing Multiple Sclerosis (RMS) Treated with Ofatumumab (KYRIOS)	Phase 4	34
8	NCT04047628	Best Available Therapy Versus Autologous Hematopoetic Stem Cell Transplant for Multiple Sclerosis (BEAT-MS)	Phase 3	156
9	NCT05084638	Study to Assess the Effect of Ofatumumab in Treatment Naive, Very Early RRMS Patients Benchmarked Against Healthy Controls (AGNOS)	Phase 4	168
10	NCT05090371	A Multicenter Study of Continued Current Therapy vs. Transition to Ofatumumab After Neurofilament (NfL) Elevation (SOSTOS)	Phase 4	150
11	NCT04926818	Efficacy and Safety of Ofatumumab and Siponimod Compared to Fingolimod in Pediatric Patients with Multiple Sclerosis (NEOS)	Phase 3	180
12	NCT04788615	Open Label Randomized Multicenter to Assess Efficacy & Tolerability of Ofatumumab 20 mg vs. First Line DMT in RMS (STHENOS)	Phase 3	236
13	NCT03500328	Traditional Versus Early Aggressive Therapy for Multiple Sclerosis Trial (TREAT-MS)	Not Applicable	900
14	NCT03535298	Determining the Effectiveness of Early Intensive Versus Escalation Approaches for RRMS (DELIVER-MS)	Phase 4	800

## Data Availability

All data utilized for the purpose of this study are available publicly and online.

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
