# Peer review of "What Do We Know So Far about Ofatumumab for Relapsing Multiple Sclerosis? A Meta-Analytical Study"

_healthcare, 2022, doi:10.3390/healthcare10112199_

Round 1

Reviewer 1 Report

This manuscript provided a brief summary of the current knowledge about Ofatumumab for relapsing multiple sclerosis. According to the authors analysis, the study showed favorable findings of Gd+ T1 lesions and the relapse rate of MS was also reduced post ofatumumab administration. As Ofatumumab is a newly approved medicine, this study is valuable by providing with a preliminary understanding of the medicine effectiveness.

However, this manuscript seems referred the same clinical trials with the study of Connie Kang et al., 2022 (Ofatumumab: A Review in Relapsing Forms of Multiple Sclerosis), in which they also summarized this medicine could reduce the Gd-enhancing T1 lesions and the relapse. Can the authors clarify the novel points of this study?

 Besides, the authors should pay attention to these points:

1.      Abbreviations of the draft. For example, in the abstract, “MS” and “Gd+T1” should be changed into the full names.

2.      In the introduction, line 31-33. The authors should provide more detailed information of the properties. For example, what are the properties of Ofatumumab are, in repleting the B cells; or could the medicine elicit high or low anti-drug responses?

3.      In line 38, actually the authors overstated the significance of this study by the word “explore”. Please rewords it.

Author Response

To Reviewer 1, thank you for taking out valuable time and reviewing this study. Below, I have attached your comments and responses. 

Reviewer 1 Comment 1:

This manuscript provided a brief summary of the current knowledge about Ofatumumab for relapsing multiple sclerosis. According to the authors analysis, the study showed favorable findings of Gd+ T1 lesions and the relapse rate of MS was also reduced post ofatumumab administration. As Ofatumumab is a newly approved medicine, this study is valuable by providing with a preliminary understanding of the medicine effectiveness.

Author Response to Comment 1: 

Respected reviewer, thank you for your favorable feedback.

Reviewer 1 Comment 2:

However, this manuscript seems referred the same clinical trials with the study of Connie Kang et al., 2022 (Ofatumumab: A Review in Relapsing Forms of Multiple Sclerosis), in which they also summarized this medicine could reduce the Gd-enhancing T1 lesions and the relapse. Can the authors clarify the novel points of this study?

Author Response to Comment 2: 

Thank you for doing your due diligence. If you noticed, the review you mention was already cited in our study. On reviewing it in full, there are absolutely no similarities to our paper. The paper you cite is not a meta-analysis. It is more of a narrative review coupled with post-hoc analysis. If you look closely, other than the trials you mention, which is a current topic, nearly 100% of the paper has no linkage to it. Our agenda/outcomes/presentation/content is poles apart. I hope that helps in clarifying. 

Reviewer 1 Comment 3:

Besides, the authors should pay attention to these points:

Abbreviations of the draft. For example, in the abstract, “MS” and “Gd+T1” should be changed into the full names.

Author Response to Comment 3: 

Thank you for noting and enlisting these issues. These inconsistencies have been fixed throughout the paper. Changes are highlighted in yellow.

Reviewer 1 Comment 4:

In the introduction, line 31-33. The authors should provide more detailed information of the properties. For example, what are the properties of Ofatumumab are, in repleting the B cells; or could the medicine elicit high or low anti-drug responses?

Author Response to Comment 4: 

In line with your comment, significant changes/additions have been made. These are highlighted in yellow.

Reviewer 1 Comment 5:

In line 38, actually the authors overstated the significance of this study by the word “explore”. Please reword it.

Author Response to Comment 5: 

Your comment is appreciated. It has been reworded. The changes are highlighted in yellow.

Regards,

ZS

Reviewer 2 Report

Thank you for submitting the important work to the journal. Here are my few comments on the manuscript: 

1) Introduction: can you add elaborate more on objectives of study while stating your hypothesis. 

2) Methods: Inclusion, exclusion criteria is missing, period of data base search is missing, study quality assessment criteria is missing, data synthesis section is missing. Please go through systematic details of these sections which are important for meta-analysis.

3) Results: please edit table 1- in making it more clear and concise. If you need to distribute table 1 into two tables to make it more clear, then please do so. 

Please edit your manuscript based on the comments above.

Author Response

Dear Reviewer 2, thank you for taking out valuable time and reviewing this study. Below, I have attached your comments and responses. Thank you.

Reviewer 2:

Thank you for submitting the important work to the journal. Here are my few comments on the manuscript: 

Reviewer 2 Comment 1:

Introduction: can you add elaborate more on objectives of study while stating your hypothesis. 

Author Response to Comment 1:

The introduction has been updated to add the objectives and more supporting data to develop and enhance the introduction. The changes are highlighted in yellow.

Reviewer 2 Comment 2: 

Methods: Inclusion, exclusion criteria is missing, period of database search is missing, study quality assessment criteria is missing, data synthesis section is missing. Please go through systematic details of these sections which are important for meta-analysis.

Author Response to Comment 2:

Respected reviewer, the methods have been fully updated to ensure your comment is accounted for; the changes are highlighted in yellow.

Reviewer 2 Comment 3:

Results: please edit table 1- in making it more clear and concise. If you need to distribute table 1 into two tables to make it more clear, then please do so. 

Please edit your manuscript based on the comments above.

Author Response to Comment 3:

Respected reviewer, the table has been broken down into two tables for clarity. Have a look.

Regards, 

ZS

Round 2

Reviewer 1 Report

Most of my points were addressed by the authors. But it seems that they did not make any change in the new version line 40-41 (line 31-33 in old version), despite the authors replied as "Author Response to Comment 4: In line with your comment, significant changes/additions have been made. These are highlighted in yellow." 

Author Response

Dear Reviewer,

The additions are made before new lines 40-41. 

Please review the additions as follows:

-B cell lysis associated with Ofatumumab correlates to completement-dependent cytotoxicity and antibody-dependent cell-mediated cytotoxicity [4]; this is unlike the mechanism see in other treatments including Ocrelizumab and Rituximab, which only bind to the large extracellular loops of the CD20 antigen; Ofatumumab binds to both the small and large extracellular loops [4,5]. 

and 

-In MS, B cells are posited to act via the antibody production and antigen presentation system to activate T cells; they are also a vital source of pro-inflammatory cytokines, which in unison orchestrate inflammatory infiltration in the central nervous system [4]. 

These changes are pasted above were made in response to your comment. 

Please let me know if that is clear.

Regards